# CSO: Constraint-guided Space Optimization for Active Scene Mapping

## ABSTRACT

Simultaneously mapping and exploring a complex unknown scene is an NP-hard problem, which is still challenging with the rapid development of deep learning techniques. We present CSO, a deep reinforcement learning-based framework for efficient active scene mapping. Constraint-guided space optimization is adopted for both state and critic space to reduce the difficulty of finding the global optimal explore path and avoid long-distance round trips while exploring. We first take the frontiers-based entropy as the input constraint with the raw observation into the network, which guides the training start from imitating the local greedy searching. However, the entropy-based optimization can easily get stuck with few local optimal or cause inefficient round trips since the entropy space and the real world do not share the same metric. Inspired by constrained reinforcement learning, we then introduce an action mask-based optimization constraint to align the metric of these two spaces. Exploration optimization in aligned spaces can avoid long-distance round trips more effectively. We evaluate our method with a ground robot in 29 complex indoor scenes with different scales. Our method can perform 19.16% more exploration efficiency and 3.12% more exploration completeness on average compared to the state-of-the-art alternatives. We also implement our method in real-world scenes that can efficiently explore an area of 649 $m^2$. The experiment video can be found in the supplementary material.

## KEYWORDS

Active Mapping, Space Alignment, Constrained Reinforcement Learning, Information Entropy, Graph Neural Network.

## 1 INTRODUCTION

Building a complete map from an unknown indoor environment based on robot scanning is critical for many applications in computer vision and robotic communities. However, it is time-consuming and inconvenient if the scanning is human-operated. Introducing artificial intelligence to guide the robot scanning automatically is then proposed by many previous works[9, 16, 17, 27, 42, 43, 45], which form the optimization problem of active scene mapping.

The goal of active mapping is finding the shortest exploration path to perform the complete scanning. However, solutions based on global planning theory, like the Travelling Salesman Problem (TSP) [30], formulate it as an NP-hard problem[26], which makes

**Unpublished working draft. Not for distribution.**

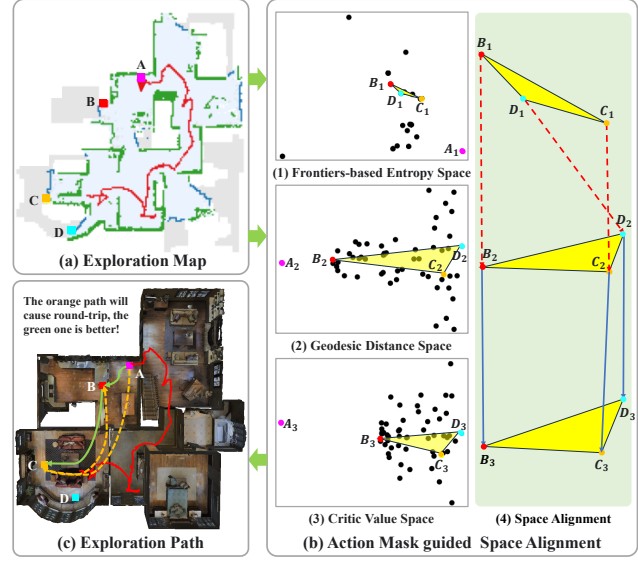

(a) Exploration Map

(1) Frontiers-based Entropy Space

(2) Geodesic Distance Space

(3) Critic Value Space

(4) Space Alignment

(b) Action Mask guided Space Alignment

The orange path will cause round-trip, the green one is better!

(c) Exploration Path

**Figure 1: Illustration of space alignment for avoiding long-distance round trips. We employ Multidimensional Scaling (MDS) to visualize the metrics of different spaces in (1), (2), and (3), where points $A$, $B$, $C$, and $D$ represent the robot's position, the optimal position, a position with higher frontier entropy, and a position farther from the robot, respectively. When two metric spaces are aligned, the distances between any two points in these spaces should be similar or exhibit proportional scaling. As shown in (4), we select $B_i, C_i, D_i$ from spaces (1), (2), (3) to construct 3 triangles $\triangle B_i C_i D_i, i = 1, 2, 3$ individually. It is evident that $\frac{B_2 C_2}{B_3 C_3} \approx \frac{C_2 D_2}{C_3 D_3} \approx \frac{B_2 D_2}{B_3 D_3}$, yet $\frac{B_1 C_1}{B_2 C_2} \not\approx \frac{C_1 D_1}{C_2 D_2} \not\approx \frac{B_1 D_1}{B_2 D_2}$, which suggests that space (3) aligns with (2) but not with (1). The final result is depicted in (c), the robot prefers the closer point $B$ with lower entropy instead of the farther point $C$ with higher entropy, thus avoiding long-distance round trips(orange dashed path, $A \to C \to B$) and follow a more rational path (green solid path, $A \to B \to C$).**

it unsolvable if the environment is large and complex. Hybrid-stage optimization[6, 34], which combines the greedy search and global planning, is then proposed for map reconstruction of large scenes. The core idea is that global planning is only adopted at the coarse-grained level of the mapping, while explicit local exploring is performed by greedy search based on information entropy. This solution shows good time efficiency even if the unknown environment is complex. However, the two optimization stages are relatively independent. Local exploration sometimes would drive global planning into a local optimal.

The recently developed reinforcement learning techniques provide another efficient solution for this NP-hard problem. A critic

network is introduced to replace the handcrafted rules to estimate the importance of each exploration destination. A correlated action network is adopted to perform the final planning. [43] takes advantage of the recent reinforcement learning solutions and neural graph networks for more efficient and complete map construction with multi-robots, whose excellence is that the whole optimization is end-to-end. However, raw scanning data is not informative enough if we only have a single robot for mapping. Bad mapping initials and noised input without cross-validation between robots highly limit the performance of previous multi-robot-based methods, which sometimes leads to long-distance round trips during global planning in a complex scene.

Additionally, the misalignment of space metric between the environment and corresponding critic value space leads to random stuck under exceptional circumstances since the robot chooses the next exploration goal based on the critic value estimated for each position. More specifically, the misalignment would make goal positions with similar critic values distribute around the whole scene. This situation would keep the robot on the run with round trips between these locations. Therefore, aligning the metric between these two spaces, which gives the critic network more specific guidance, is critical for efficient active mapping with a single robot.

This paper proposes a novel reinforcement learning-based active mapping approach with constraint-guided space optimization for both state and critic space. Specifically, we first introduce a frontiers-based entropy as the input constraint with the raw observation into the network to boost the local optimization. The advantage of this is that the heuristic-based entropy can cover the bad planning given by the critic network with incomplete initials, which can significantly enhance the mapping efficiency at the beginning. It is important to emphasize that this extra input would not mess up the global planning with enough observation like the previous hybrid-stage optimization approaches. Since the entropy is given implicitly through the network, the two-stage optimizations are highly correlated.

The second stage of space optimization is performed by the action network, which takes the outputs of the critic network to give the global planning. We introduce an action mask strategy to solve the misalignment problem of the space metric we mentioned above. It is not easy to align the metric without explicit supervision. Inspired by constrained reinforcement learning[21, 23, 44], we utilize the relevance of the critic network and action network to perform the alignment. At the beginning of training, we constrain the action network through a local action mask with a limited size near the robot. In response, the critic network tends to tune down the estimated scores for these long-distance goals since the action network is not selecting them. And then the critic value space will gradually align with the real-world space, as shown in figure 1.

We evaluate our method with a ground robot in 29 complex indoor scenes. In the experiments, our method can perform 19.16% more exploration efficiency and 3.12% more exploration completeness on average compared to the state-of-the-art alternatives. Furthermore, the experiments demonstrate that our method also has significantly better mapping performance on more complex and larger-scale scenes thanks to the frontiers-based entropy design. Some visual examples are given in figure 6, which demonstrate the proposed action mask's effectiveness in avoiding long-distance

round trips during active mapping. We also implement our method with a ground robot in 3 real-world scenes that can efficiently explore areas from 170 $m^2$ to 649 $m^2$, as shown in figure 7. In summary, the contributions of this paper include the following:

- We introduce a frontiers-based entropy to constrain the network input, which can significantly improve training efficiency and mapping performance. This design explored a novel way to make integration of the heuristic-based method and advanced reinforcement learning-based techniques.
- We propose an action mask constraint to guide the network to do efficient global planning by aligning the metric between the critic value space and the real-world space. This design is demonstrated to be effective for avoiding round trips during the exploration, which can significantly improve the efficiency of active mapping.
- The proposed constraint-guided space optimization-based active scene mapping outperforms state-of-the-art alternatives with indoor scenes, which gains 19.16% more exploration efficiency and 3.12% more exploration completeness on the Matterport3D dataset.

## 2 RELATED WORK

**Traditional Heuristic Method:** The pioneering work [42] first proposed the concept of the frontier for active mapping, i.e., the boundary areas between explored free space and unexplored space, aiming to guide the robot to the frontiers until the entire space is observed. Many subsequent works [2, 11, 13, 17, 20, 37, 46, 47] achieve both 2D and 3D active mapping based on this concept. In addition, [6, 16, 19] have also achieved good performance based on Rapidly Exploring Random-Trees [24], Travelling Salesman Problem [30], etc. These methods work well for active mapping. However, most of them either greedily select frontiers, making it difficult to obtain an approximate global optimal solution, or the planning process is time-consuming and computationally expensive.

**Learning-Based Method.** [9, 10, 43] employ convolution neural network or graph neural network [22] to regress the goal position by maximizing the long-term value via reinforcement learning, providing less planning time and approximate optimal strategy. These works directly use raw data as state input without further data processing, which will improve exploration efficiency. In addition, [7] successfully uses an attention-based deep reinforcement learning approach for active mapping. However, they only conduct experiments in perfect and synthetic 2D scene maps, so for realistic scene data such as Matterport3D [8], Gibson [41], and real-world, it may cause the performance drop due to the lack of sim-to-real.

**Information-Theoretic Methods** Another popular way for active mapping is using the information theory. By calculating the information gain or uncertainty value [3, 14, 15, 19, 29, 31, 32], the robot chooses actions that can maximize information gain or reduce uncertainty of the scene. The earliest information-theoretic strategies are those proposed by [40] and [18]. [1, 4, 38, 39] maximize information gain over the next few actions. And these works can also be extended to solve multi-robot tasks [5, 12, 28].

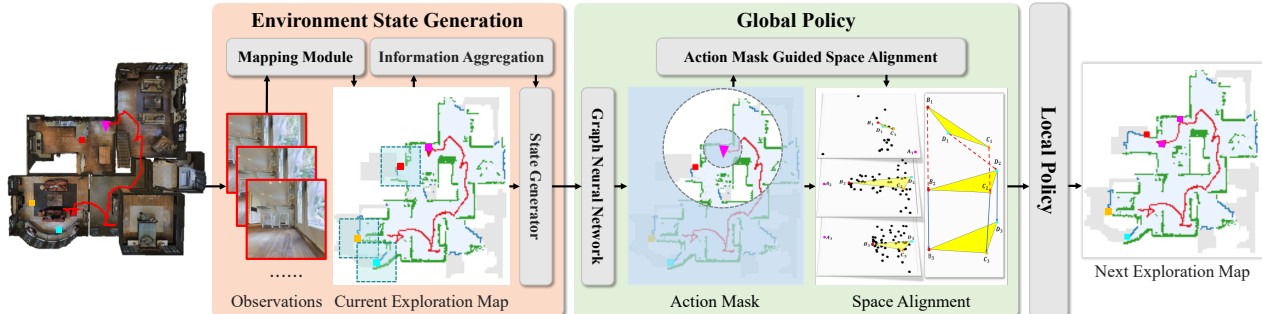

**Figure 2: The Overview of CSO for Active Scene Mapping. The robot first uses the Mapping Module to construct an Exploration Map based on the Observations $C(\omega_t)$, and uses Information Aggregation to compute the frontiers-based entropy $I(M_t)$ of each frontier(Section 3.2). Then the State Generator generates the state input $s(\omega_t)$ for the Global Policy. At the global planning stage, the Graph Neural Network-based encoder is used for feature fusion and extraction(Section 3.4). Based on the sampling probabilities of frontiers output by the actor network, the Action Mask Guided Space Alignment(Section 3.3) filters unreasonable frontiers and guides the critic value space to align with the geodesic distance space. Once a frontier is selected as the long-term goal, the Local Policy drives the robot to reach this goal and update the Exploration Map based on new observations $C(\omega_{t+1})$. This planning cycle is iteratively implemented until the termination criteria are triggered.**

## 3 METHOD

### 3.1 Problem Statement and Formulation

The goal of active mapping is to let the robot automatically decide its next action based on its local observations in an unknown environment and gradually generate a globally optimal shortest exploration path, which can completely scan the scene and build a complete map. This optimization process can be defined as:

$$L = \min_{\Omega = \{\omega_0, \omega_1, \dots, \omega_k\}} \sum_t \text{dist}(\omega_t, \omega_{t+1}), \qquad (1)$$

$$s.t. \left| E - \sum_t C(\omega_t) \right| = 0$$

where $L$ is the exploration trajectory length, $\text{dist}(\cdot)$ is the travel length from location $\omega_t$ to $\omega_{t-1}$, $C(\cdot)$ is the coverage area from a certain scanning location, $E$ is the complete environment mapping, $\omega_t$ is the robot's location at $t$. Note that $E$ and $C(\cdot)$ are unknown before the mapping. Thus this optimization can not be solved directly.

Since we can not solve the optimization at one step, we formulate equation 1 as a Markov Decision Process (MDP) to solve this ill-posed problem step-by-step, whose training details can be found in section 3.4. Specifically, the learning process is determined by the tuple: $(S, A, \Pi, R)$, where $S$ is the state space constructed from the scanning during the exploration while $A$ is the action space of the robot. Note that we use the frontiers of the occupancy map constructed while exploring the scene as the state and action space, which can greatly reduce the space size and make the learning process easier to converge. Furthermore, we propose a frontiers-based entropy as the input constraint for the state space $S$, which guides the training start from imitating the local greedy searching. Details about $S$ can be found in section 3.2. We then introduce an action mask design as a constraint to align the metric of $S$ and the critic space given by $\Pi$. $\Pi(a|s)$ denoting the probability of selecting

action $a$ under state $s$. Details of action mask design can be found in section 3.3. The overview of our pipeline is shown in figure 2.

### 3.2 Environment State Design

The state for our framework mainly consists of three parts, the occupancy map $M_t$, the distance map $D(M_t, \omega_t)$, and the proposed frontiers-based entropy $I(M_t)$.

$$s(\omega_t) = (M_t, D(M_t, \omega_t), I(M_t)) \qquad (2)$$

*Occupancy map.* Given the observation $C(\omega_t)$ from the robot at location $\omega_t$, a 2D global map from the top-down view of the 3D scene is constructed as the occupancy map. The occupancy map is denoted as $M_t \in [0, 1]^{X \times Y \times 2}$ at time step $t$, where $X, Y$ are the map size. The two channels indicate the explored and occupied regions separately. Each cell in $M_t$ can be classified as one of the three classes, open (explored but not occupied), occupied, and unknown (unexplored). Frontier cells $F_t \subset M_t$ are defined as the open cells whose adjacent to at least one unknown cell.

*Distance map.* Given the current position $\omega_t$ and the constructed occupancy map $M_t$ build by the robot while exploring, we further construct a distance map $D(M_t, \omega_t) \in \mathbb{R}^{X \times Y}$, in which $D_{x,y}(M_t, \omega_t)$ represents the geodesic distance from the location $(x, y)$ to the robot's position $\omega_t$.

$$D_{x,y}(M_t, \omega_t) = \text{dist}^{M_t}((x, y), \omega_t) \qquad (3)$$

The geodesic distance $\text{dist}^{M_t}$ is the shortest distance for traversal between two points in $M_t$ without collision. Compared with the Euclidean distance, it can implicitly encode the underlying scene layout information, which gives better guidance for robots to explore. We compute $\text{dist}^{M_t}$ with the Fast Marching Method [35].

*Frontiers-based Entropy.* Besides the raw data $M_t$ and $D(M_t, \omega_t)$, we would like to introduce some high-level information as a constraint to reduce the searching space while the $M_t$ is highly incomplete. Heuristically, the frontier cell $f \in M_t$ indicates the potential

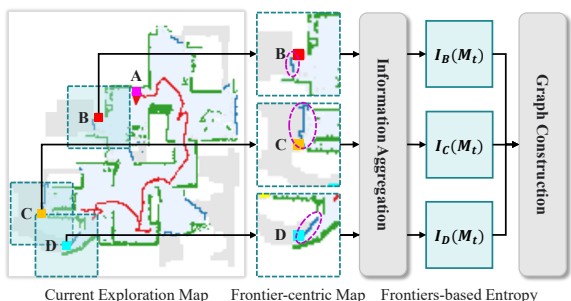

Current Exploration Map    Frontier-centric Map    Frontiers-based Entropy

**Figure 3: We use the information aggregation module to compute the frontiers-based entropy based on the frontier-centric map, which is a $\gamma \times \gamma$ submap of the exploration map centered on each frontier. The dark blue pixels in the purple dashed circles denote the frontiers.**

next best goal to explore since its surrounding is information incomplete. And the frontiers in a scene usually do not exist alone, but have the characteristics of small-scale aggregation and large-scale dispersion. Inspired by this, we propose a frontiers-based entropy $I$ to encode such information as input to constrain the network searching, which can be described as:

$$I_{x,y}(M_t) = \| \{ f \in F_t | \text{dist}^{M_t}(f, (x, y)) < \gamma \} \| \tag{4}$$

where $I_{x,y}(M_t)$ denotes the number of frontiers calculated by the Information Aggregation module based on the $\gamma \times \gamma$ submap centered on $(x, y)$ in $M_t$, as shown in figure 3. By doing this, the information contained in each location, in addition to its map coordinates $(x, y)$, also contains statistical information within its neighborhood. Note that $I_{x,y}(M_t)$ will give $(x, y)$ a larger information entropy if it is surrounded by more frontiers. For the discrete frontiers or noise points, a smaller information entropy can appropriately reduce its impact on global optimization.

## 3.3 Action Mask guided Space Alignment

In the reinforcement learning framework, the robot chooses the next exploration goal $\omega_{t+1}$ based on $\omega_{t+1} = \arg\max_{f \in F_t} \Pi(a|s(\omega_t))$. The action space $A$ is reduced to selecting a frontier $f$ from $F_t$. If the world space $M_t$ is not aligned with the space metric of $\Pi(a|s(\omega_t))$, it will cause the robot to randomly get stuck or turn back and forth under exceptional circumstances, as shown in figure 1.

Therefore, aligning the metric between $M_t$ and $\Pi(a|s(\omega_t))$, which gives the critic network more specific guidance, is critical for efficient active mapping with a single robot. Since there is no additional supervision information, we introduce an action mask strategy to solve the misalignment problem of the space metric. Specifically, we design two action masks: the valid distance mask and the stuck mask, to filter the actions in the action space of the global policy, and constrain action sampling to be performed within the valid action space, rather than the entire action space. By doing this, the influence of abnormal data in the network learning process can be reduced, and the network learning rate can be accelerated at the same time so that $M_t$ and $\Pi(a|s(\omega_t))$ can be aligned.

*Valid Distance Mask.* We first design a valid distance mask to filter some obviously invalid goals in the action space. When the

robot explores a large-scale complex scene, a reasonable and efficient exploration method should make the robot gradually explore the scene instead of wandering back and forth between multiple goals. We want the better actions given by $\Pi(a|s(\omega_t))$ distributed in a reasonable range around $\omega_t$ to ensure the gradual exploration. This heuristic property also implicitly requires the metric of $M_t$ and $\Pi(a|s(\omega_t))$ to be aligned.

To ensure the constraint is appropriate, not too strict or loose, we filter the action space based on the geodesic distance information from $\omega_t$ to each potential goal. Specifically, we set 2 thresholds, $\beta_{\text{near}}$ and $\beta_{\text{far}}$. For the potential goals that exceed the threshold range $[\beta_{\text{near}}, \beta_{\text{far}}]$ of $\omega_t$, we set their sampling probability to 0 to ensure that they will not be sampled, and only sample frontiers within the valid distance. The masked $\Pi_{\text{mask}}(f|s(\omega_t))$ is given as:

$$\Pi_{\text{mask}}(f|s(\omega_t)) = \begin{cases} \Pi(f|s(\omega_t)), & \text{if } dist^{M_t}(f, \omega_t) \in [\beta_{\text{near}}, \beta_{\text{far}}] \\ 0, & \text{otherwise} \end{cases} \tag{5}$$

where $\Pi_{\text{mask}}(f|s(\omega_t))$ represents the masked probability of the action to select $f$ as the next exploration goal, $dist^{M_t}(f, \omega_t)$ is the geodesic distance from robot position $\omega_t$ to the $f$, and $\Pi(f|s(\omega_t))$ is the original probability distribution from the network. We set $\beta_{\text{near}} = 0.2$ and $\beta_{\text{far}} = 2$ in our experiments. When there are no valid distance frontiers within the range $[\beta_{\text{near}}, \beta_{\text{far}}]$, we first remove the mask of $\beta_{\text{near}}$ to extend the valid range to $[0, \beta_{\text{far}}]$. If there are still no valid distance frontiers within range $[0, \beta_{\text{far}}]$, then we further remove the mask of $\beta_{\text{far}}$ and select the goal from all frontiers.

---

**Algorithm 1:** Stuck Mask Filtering.

**Input:** Action function $\Pi_{\text{mask}}(a|s(\omega_t)$. Past 3 selected exploration goals $\{\omega_t, \omega_{t-1}, \omega_{t-2}\}$ and corresponded exploration area $\{\|M_t\|, \|M_{t-1}\|, \|M_{t-2}\|\}$.

**Output:** Stuck masked $\Pi_{\text{mask}}(a|s(\omega_t)$

Calculate the max moving length in the last 3 steps:
  $l_{\max} = \max_{i \in \{t-1, t-2\}} dist^{M_t}(\omega_i, \omega_{i+1})$
Calculate the max area increment in the last 3 steps:
  $c_{\max} = \max_{i \in \{t-1, t-2\}} \|M_i - M_{i+1}\|$
Given 3 thresholds $\alpha_1 = 1$, $\alpha_2 = 1$ and $\alpha_3 = 1$
**if** $l_{\max} < \alpha_1$ and $c_{\max} < \alpha_2$ **then**
  **for** Location $\omega_a$ given by $a$:
    $\min_{i \in \{t, t-1, t-2\}} dist^{M_t}(\omega_i, \omega_a) < \alpha_3$ **do**
    |   $\Pi_{\text{mask}}(a|s(\omega_t) = 0$
  **end**
**end**

---

*Stuck Mask.* Besides the valid distance mask, we further design a stuck mask, which is used to filter the actions from the action space that will cause the robot to continue being stuck. To be specific, we first determine whether the robot is stuck by collecting the robot's long-term goals, exploration areas, and moving distances in the past 3 time steps. We consider the robot to be stuck if the maximum moving length is less than the threshold $\alpha_1 = 1$ and the maximum area increment is less than the threshold $\alpha_2 = 1$. Then, we calculate the distance from the candidate actions to the past 3 long-term goals. Only when the minimum distance is greater than the threshold $\alpha_3 = 1$, we consider the action to be a reasonable action that has a probability of being selected as the long-term goal. By doing this, a

new long-term goal that is different from the historical goals will be chosen, which can help the robot get out of stuck. Details of the stuck mask operation can be found in Algorithm 1.

## 3.4 Deep Reinforcement Learning Framework

We use the off-policy learning approach Proximal Policy Optimization (PPO) [33] as the policy optimizer. The actor is trained to learn a policy network that outputs the sampling probability of each action. Graph neural network [22] is adopted as the encoder to perform feature extraction and fusion. The critic trains a state-value network that predicts the state value $V(s(\omega_t))$ to indicate how much reward is earned from the current state, which is adopted to train our actor network. Details of the network can be found in the supplementary materials.

*Graph Neural Network-based Encoder.* We first construct a graph $G(F_t, \Omega_t)$ based on the frontiers $F_t$ and the explored path $\Omega_t = \{\omega_0, ..., \omega_t\}$ to represent the context of the current scene. It establishes the correspondences between the robot and frontier nodes extracted from the constructed occupancy map $M_t$. We distribute the information given by state $s(\omega_t)$ into nodes and edges of $G(F_t, \omega_t)$. For each node $n_i$, the input feature $f(n_i) \in \mathbb{R}^5$ includes the $(x, y)$ coordinates in $M_t$, the semantic label that indicates $n_i \in F_t$ or $n_i \in \Omega_t$, the history label that indicates $n_i$ is $\omega_t$ or explored nodes $n_i \in \{\omega_0, ..., \omega_{t-1}\}$ and the proposed frontiers-based entropy $I_{n_i}(M_t)$. The edge feature $f(n_i, n_j) \in \mathbb{R}^{32}$ is given by a multi-layer perception(MLP) with $l_{ij} \in \mathbb{R}^1$ indicates the geodesic distance from node $n_j$ to node $n_i$. Then we feed these node and edge features into the GNN network, which is our actor network, for feature transfer and output a set of scores. Based on these scores, we compute the probability $\Pi_{mask}(f|s(\omega_t))$ of each action, followed by action mask-based action sampling.

*Reward.* Our training goal is to maximize the accumulated reward function $G(\omega_t|\Omega_{t-1})$. For the active mapping task, the goal is to pursue high-time efficiency and map completeness. To achieve this goal, we use an efficiency reward $R_{step}$ and a coverage reward $R^t_{coverage}$. The efficiency reward $R_{step}$ punishes unnecessary time steps to encourage the robot for more efficient exploration. And the coverage reward $R^t_{coverage} = \|M_t\| - \|M_{t-1}\|$ is defined as the coverage increment at time step $t$, where the coverage $\|M_t\|$ is the area of the open space in the occupancy map $M_t$. Then the complete reward function formula is defined as

$$G(\omega_t|\Omega_{t-1}) = \sum_{t=0}^{t-1} (R_{step} + \lambda_c R^t_{coverage}) \qquad (6)$$

where $\lambda_c$ is the hyper-parameter to balance these two rewards.

## 4 EXPERIMENT

### 4.1 Experimental Setup

We test our method in both simulated and real-world environments. For experiments in simulation, our experiments are conducted on the Matterport3D [8] dataset using the iGibson[25, 36] simulator. For experiments in the real world, we deploy a LIMO robot to explore 3 distinct real-world scenes, as shown in figure 7. Detailed

implementations and the video of real-world experiments can be found in the supplementary material.

*Data processing.* We conducted simulation experiments on the Matterport3D dataset [8] containing 90 realistic indoor scenes using the iGibson simulator [25, 36]. We followed the standard train/val/test split of Matterport3D and further split them into four scales: small($< 30m^2$), middle($30 - 100m^2$), large($100 - 260m^2$) and super large($> 260m^2$) according to their traversable areas. Details of the splits can be found in the supplementary materials.

*Termination criteria.* We consider exploration complete when the coverage is greater than 99% while the unexplored area is less than $1m^2$, or there are no accessible frontiers in the environment. Meanwhile, we also set the maximum exploration steps as $n = 3000$. If the exploration step exceeds $n$, we terminate exploration anyway.

*Evaluation metrics.* We evaluate the map completeness via the coverage(%) that calculates the percentage of explored open space over the ground truth open space in the environment, and the exploration area($m^2$) that calculates the area of explored open space in the environment. In addition, we use the ratio of exploration area to the time($step$) as the metric to evaluate the scene exploration efficiency($m^2/step$), which represents the average area scanned at each step while exploring a scene.

## 4.2 Alternatives for Comparison

We compare our method with learning-based methods (ANS, UPEN, ARIADNE, NeuralCoMapping) and traditional heuristics methods (Greedy-Info, Greedy-Dist), details of each method are as follows:

- **Active Neural SLAM(ANS) [9].** ANS learns a policy that uses an egocentric local map and a geocentric global map as input and regresses the goal estimation for path planning.
- **UPEN [19].** UPEN learns the occupancy priors over indoor maps to generate occupancy maps beyond the field-of-view of the agent and then leverages the model uncertainty over the generated areas to formulate path selection policies for the task of interest. In this work, the Rapidly Exploring Random-Trees(RRT) [24] is used for path planning.
- **ARIADNE [7].** ARiADNE proposes a reinforcement learning approach that relies on attention-based deep neural networks for autonomous exploration. However, they only conduct experiments in perfect and synthetic 2D scene maps instead of realistic 3D simulated scenes such as Matterport3D[8]. Therefore, we pre-generate 2D maps of all scenes with the same initialization information and then conduct tests.
- **NeuralCoMapping [43].** Following the recent graph neural-network-based reinforcement learning solution NeuralCoMapping, we adapt their work from a multi-robot active mapping to a single-robot for comparison with our approach.
- **Greedy-Info and Greedy-Dist [42].** Based on the frontiers proposed by [42], we further design two heuristics methods, Greedy-Info and Greedy-Dist, as two variants of our method and use them as two alternatives for comparison based on the greedy strategies. These two methods choose the long-term goal in a greedy manner based on the geodesic distance or frontiers-based entropy. Detailed implementations can be found in the supplementary material.

Table 1: Comparison with alternatives on the Matterport3D dataset [8].

| Method | Exploration Area($m^2$) | Coverage(%) | Time($step$) | Efficiency ($m^2/step$) |
|---|---|---|---|---|
| ANS [9] | 50.36 | 57.83 | 2754.81 | 0.02096 |
| UPEN [19] | 39.63 | 41.09 | 3000.00 | 0.01321 |
| ARIADNE [7] | 70.80 | 72.37 | 2601.03 | 0.02665 |
| NeuralCoMapping [43] | 73.44 | 72.67 | 2108.74 | 0.04092 |
| Greedy-Info [42] | 50.38 | 48.73 | 2707.20 | 0.02284 |
| Greedy-Dist [42] | 73.86 | 73.13 | 2185.47 | 0.03890 |
| Ours | **81.34** | **75.79** | **2037.77** | **0.04876** |

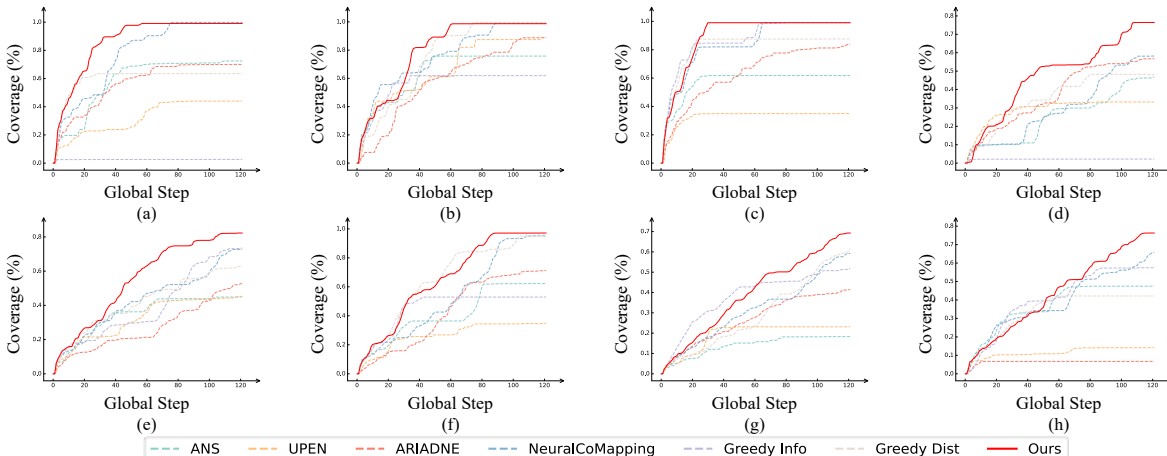

Figure 4: We compared the coverage curves over global steps in 8 Matterport3D scenes with different scales. The closer the curve is to the upper left corner, the corresponding method can explore a larger area over the same time, indicating a more efficient exploration. It is obvious that our method(solid) is closer to the upper left than others(dashed).

## 4.3 Evaluation

*Quantitative Comparison with State-of-the-art Alternatives.* We compared our algorithm with all alternatives in 29 prior unknown scenes consisting of the Matterport3D test and val dataset according to the standard split. We conducted 5 tests for each scene, with all methods starting with the same position and orientation, and the final statistics are shown in table 1. It's obvious that our method outperforms all alternatives in terms of exploration completeness, time steps, and efficiency, demonstrating the superiority of our proposed method. Specifically, our method improves the exploration efficiency by about 19.16% and coverage by about 3.12% compared with the state-of-the-art method NeuralCoMapping [43].

It can also be found from figure 5 that the superiority of our method in exploration efficiency over alternatives becomes more significant as the scene scale increases. This is because our method effectively reduces long-distance round trips while exploring and enables more efficient exploration. It is not a visible effect in small scenes, but as the scene scale increases, the effect of long-distance round trips on exploration efficiency becomes increasingly obvious. Therefore, the larger the scale of the scene, the more pronounced the superiority of our method becomes.

We also compared the planning time of all methods while exploring. Compared with the time-consuming traditional method like RRT(0.878$s$), our planning time(0.024$s$) is only 2.73% of it.

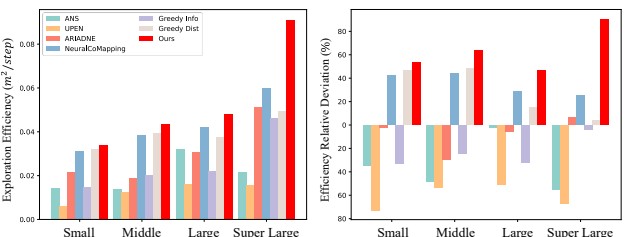

Figure 5: Comparison of exploration efficiency (top) and efficiency relative deviation (bottom) for each method across different scene scales. As the scene scale increases, the superiority of our method(red) becomes more and more obvious.

Due to using frontiers-based entropy and action mask operations, our method causes a longer planning time than other learning-based methods like NerualCoMapping(0.019$s$) and simple greedy strategies(0.015$s$). Nevertheless, our method still maintains the same level while achieving better performance. Detailed planning times for all methods can be found in the supplementary material.

*Visual Comparison with State-of-the-art Alternatives.* Figure 6 shows the visualization results of all methods in scenes with different scales from the Matterport3D dataset, where the scanning completeness and exploration paths among different alternatives

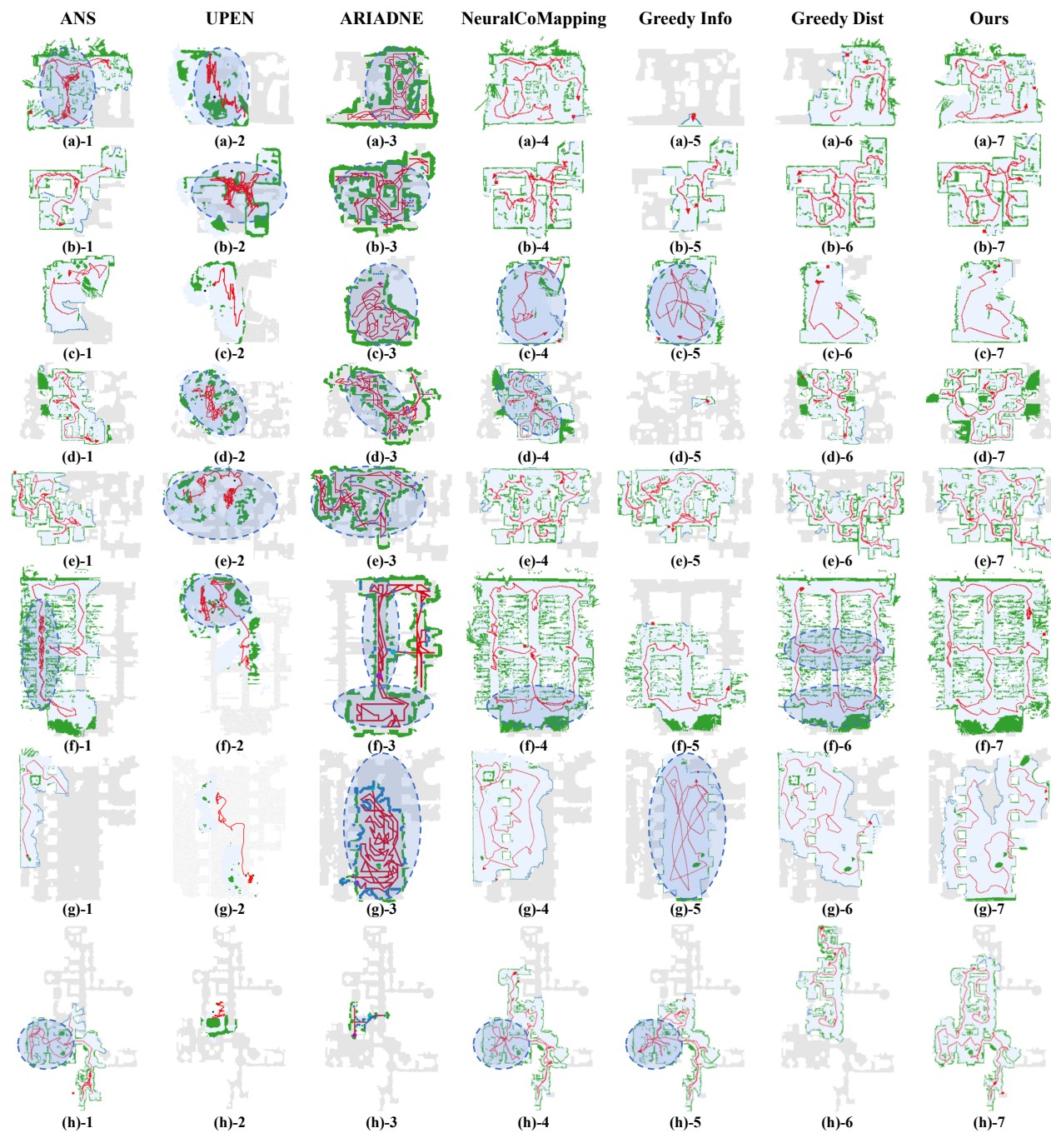

**Figure 6: Visual comparisons of our method with alternatives in the 8 scenes of figure 4, where red lines indicate the robot's exploration trajectories, gray areas are ground truth maps, light blue areas denote the explored regions, and green areas represent obstacles. We also encircle unreasonable exploration trajectories with dashed blue circles.**

can be intuitively compared. We also record the exploration coverage curves for each scene corresponding to figure 6, as shown in figure 4. It can be seen from figure 6 that our method outperforms others in both exploration coverage and path rationality. Meanwhile, our exploration curves in figure 4 are closer to the upper left, meaning our method can explore scenes efficiently.

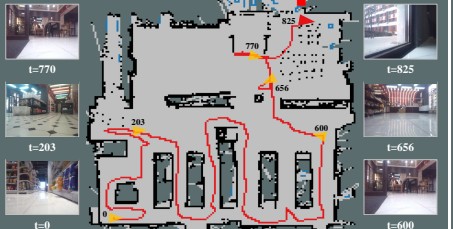
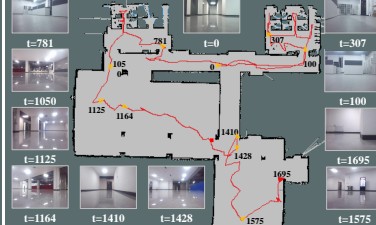

(a) Laboratories and Corridors
$176.22m^2$, 1500 $steps$

(b) Supermarket
$170.44m^2$, 825 $steps$

(c) Student Innovation and Practice Center
$649.23m^2$, 1695 $steps$

**Figure 7: We test our method in 3 real-world scenes using a LIMO robot, where the cumulative exploration area and steps are labeled below. For each exploration map, the gray areas are the explored regions, the black areas represent obstacles, the red line tracks the robot's exploration trajectory, the red triangle marks its current position and orientation, and the red square indicates its next long-term goal. Moreover, we marked some historical timesteps while exploring with the orange triangles, and the images around the exploration map are first-view RGB images of the robot at the corresponding timesteps.**

Specifically, for some small and simple scenes (a)-(c) where the whole scene can be explored within $n = 3000$ steps, our exploration paths are more reasonable and there are no long-distance round trips or serious stuck situations while exploring. As for large and complex scenes (d)-(h) that all methods cannot explore the whole scene within $n = 3000$ steps, our method can explore more regions with a more reasonable path and achieve higher exploration completeness. This is because the critic value space is aligned with the real-world geodesic distance space, preventing the robot from falling into local optimums and long-distance round trips.

In comparison, ANS [9] uses the entire map as the policy input and outputs any position in the map as the next long-term goal without any constraints, which makes it easy to select goals that will cause long-distance round trips. NeuralCoMapping [43] and ARIADNE [7] enhance exploration performance by constraining the action space to frontiers or neighboring nodes around the robot instead of the entire map. However, the same problem is still inevitable because the distance from frontiers or neighboring nodes to the robot varies across different scenes due to different scene scales and dynamics while exploring. In addition, UPEN [19] uses RRT [24] for path planning based on the occupancy map uncertainty predicted by their model and selects the path that can maximize map uncertainty over candidate paths for exploration. Therefore, it depends heavily on the model predictions. At the same time, path planning based on RRT is also time-consuming. More comparison results and discussions can be found in the supplementary material.

## 4.4 Ablation Study

We conduct ablation studies to evaluate the effects of introducing two kinds of action masks and frontiers-based entropy on exploration efficiency. The results are shown in table 2. We can find that using either action masks (+AM) or frontiers-based entropy (+Info) can improve exploration efficiency. This is because either of them can provide positive guidance for network learning, guiding the strategy to choose the actions with larger entropy value or filter out some obvious invalid actions, thus improving the exploration efficiency. However, these methods are still myopic. Especially when exploring large-scale scenes that need to maximize long-term rewards, it is still difficult to further improve the exploration efficiency by simply using either of these two approaches.

**Table 2: Ablation studies of our method, where "baseline" is the NeuralCoMapping [43], "+SM" indicates using stuck mask, "+VDM" indicates using valid distance mask, "+AM" means that both stuck mask and valid distance mask are used, and "+Info" indicates using frontiers-based entropy.**

| Method | Efficiency ($m^2/step$) |
|---|---|
| Baseline | 0.04092 |
| Baseline+SM | 0.04001 |
| Baseline+VDM | 0.03956 |
| Baseline+AM | 0.04288 |
| Baseline+Info | 0.04388 |
| Baseline+Info+SM | 0.03761 |
| Baseline+Info+VDM | 0.04333 |
| Baseline+Info+AM(Ours) | **0.04876** |

Another important thing to note is that solely employing either a stuck mask (+SM) or a valid distance mask (+VDM) will diminish exploration efficiency. Only through the combined utilization of both can exploration efficiency be enhanced. We analyze that if only one type of action masking is performed, the probability of sampling the other type of invalid actions will increase, thus leading to a decrease in exploration efficiency. Only when both action masks are used simultaneously can the policy learn effective action sequences from valid actions, thus maximizing long-term rewards and improving exploration efficiency.

## 5 CONCLUSION

We propose a constraint-guided space optimization for active scene mapping. The space optimization consists of two aspects. We first introduce the transitional heuristic rules to formulate a frontiers-based entropy to constrain the search space, which can significantly improve the exploration efficiency at the beginning of the scanning. The second constraint is introduced by our action space design, which can align the metric of the critic value space and the real-world space. This design can significantly help the robot avoid long-distance round trips while exploring a large and complex scene. The evaluation results also demonstrate the superiority of our method while comparing it with the state-of-the-art alternatives.

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
