# OpenReview forum: "CSO: Constraint-guided Space Optimization for Active Scene Mapping"
_acmmm.org/ACMMM/2024/Conference — MM2024 Poster_

### Official Review · Reviewer_xFti · 2024-05-24

**Rating:** 4
**Confidence:** 2

**Summary:**

The paper introduces a deep reinforcement learning framework to enhance active scene mapping efficiency. By employing frontiers-based entropy and action mask-based constraints, the method aligns state and critic spaces to avoid long-distance round trips. Evaluations on 29 complex indoor scenes show CSO outperforms existing methods in exploration efficiency and completeness, with additional validation in real-world settings.

**Strengths:**

1 Introduces a constraint-guided optimization approach using frontiers-based entropy and action mask-based constraints, which is innovative in the field of active scene mapping.
2 Effectively integrates deep reinforcement learning with heuristic rules to optimize state and critic spaces, ensuring better alignment and exploration paths.
3 Extensive experiments conducted in 29 complex indoor scenes using the Matterport3D dataset demonstrate a 19.16% increase in exploration efficiency and a 3.12% improvement in completeness compared to state-of-the-art methods.

**Limitations:**

1 The integration of heuristic rules such as frontiers-based entropy might limit the flexibility of the approach. Relying on such heuristics could hinder performance in scenarios where these rules are less effective.
2 The paper mentions that the planning time, although better than some traditional methods, is longer than other learning-based methods. This computational overhead might impact real-time applications where rapid decision-making is crucial.
3 While the paper performs ablation studies on action masks and entropy, it lacks a detailed analysis of the contribution of each component of the neural network, such as the Graph Neural Network and PPO components. This would help in understanding the impact of each part of the architecture.
4 The real-world experiments are conducted in a limited number of scenes. Expanding the range and variety of real-world tests would provide a more comprehensive validation of the method’s robustness and generalizability.

**Suitability:**

2

---

### Official Review · Reviewer_Xcxm · 2024-05-25

**Rating:** 4
**Confidence:** 1

**Summary:**

The paper introduces a deep reinforcement learning-based framework designed to enhance the efficiency and completeness of active scene mapping with a single robot. The framework, named CSO, employs constraint-guided space optimization by integrating a frontiers-based entropy metric and an action mask strategy. The frontiers-based entropy metric prioritizes areas with high information gain to improve local search efficiency, while the action mask strategy aligns the critic value space with the real-world space to avoid redundant long-distance movements. Evaluated in 29 complex indoor scenes and through real-world tests with a LIMO robot, CSO demonstrated a significant increase in exploration efficiency and completeness compared to state-of-the-art methods. The results highlight CSO's ability to optimize exploration paths and improve overall mapping performance in both simulated and real environments.

**Strengths:**

1. This paper is well written, and the figures are well plotted.
2. The method presents a novel architecture of how to introduce constraints based on action mask-based to improve the pipeline based on entropy optimization, which could easily fall into a local minimum and thus cause inefficiency.
3. The results adequately prove the efficiency of the method and reach state-of-the-art results.
4. Ablation studies prove the importance of each part that the authors proposed.

**Limitations:**

1. Although the methods are well introduced, some parts of intuitions are not well discussed.
2. Deep reinforcement learning methods generally require significant training time and extensive data. The paper does not discuss the training duration and the amount of data required, which could be a practical limitation for deployment.
3. I am not an expert in this field, and I will also refer to other reviewers' feedback.

**Suitability:**

2

---

### Official Review · Reviewer_kBJy · 2024-05-26

**Rating:** 3
**Confidence:** 3

**Summary:**

The paper presents a method for efficient exploration of complex unseen environments that uses an action mask-based constraint to align the entropy/metric space with the critic-value space of a reinforcement learning algorithm.
The method is composed of two stages of optimization, the first uses a frontiers-based entropy constraint, while the second uses an action mask strategy to mitigate the space misalignment between real-world metrics and critic value.

**Strengths:**

The paper is engaging and well-written, introducing an interesting method for efficiently exploring complex indoor environments using a robotic agent. The method is comprehensively detailed, ensuring that the code can be easily reproduced. The figures are seamlessly integrated with the text, enhancing the understanding of the method and offering valuable insights during the qualitative evaluation.

The experimental evaluation is comprehensive, thoroughly examining all proposed modules. This extensive evaluation demonstrates the improvement provided by the proposed approach. The qualitative experiments in Figure 6 are very interesting and useful to give an idea of the final behavior of the proposed method compared to the baselines.

The real-world deployment is an appreciable experiment to validate the performance of the method in real-world conditions.

**Limitations:**

The related work section is very brief and could be extended to better frame the proposed method in the current literature on embodied exploration methods. For example, some related work adopting learning-based exploration methods is not included in the paper [A, B]. Moreover, current literature on learning-based methods for exploration is also ignored in the experimental evaluation [31, A].

The exploration results on MP3D contained in Table 1, seem relatively low compared to other work on exploration [31] considering that the time budget is quite high (3000 steps). Can the authors elaborate on this difference?

The overall novelty of the proposed method is limited as the main contribution is given by a mask on the action space of the navigation policy.

A big concern I have about the paper is its suitability to the multimodal community as it seems there is not a clear multimodal aspect in the proposed approach, but I could be convinced otherwise. Could the authors explain how this work could be beneficial to the MM community?

Typos: line 687 -> NerualCoMapping

[A]: Bigazzi, R., Landi, F., Cascianelli, S., Baraldi, L., Cornia, M., & Cucchiara, R. (2022). Focus on impact: indoor exploration with intrinsic motivation. IEEE Robotics and Automation Letters, 7(2), 2985-2992.
[B]: Hu, Y., Geng, J., Wang, C., Keller, J., & Scherer, S. (2023). Off-policy evaluation with online adaptation for robot exploration in challenging environments. IEEE Robotics and Automation Letters.

**Suitability:**

1

---

### Meta-Review · Area_Chair_AASb · 2024-06-30

**Recommendation:** Accept (Poster)
**Confidence:** 5

**Metareview:**

This paper proposes a deep RL method for efficient active scene mapping. To tackle the problems of the few local optimal and inefficient round trips in the entropy-based optimization, the paper designs an action mask-based optimization constraint to align the metric of the entropy space and the real world. Extensive experiments in indoor scenes demonstrates its effectiveness over the previous SoTA and a real-world experiment is conducted to show its efficiency.

All three reviewers agree on the acceptance of this paper for its effectiveness shown in the experiment, and the appreciable real-world deployment. Concerns such as the insufficient related work, the lack of novelty, and some format issues are resolved in the rebuttal. After considering all these effort, AC agrees with reviewers that the paper should be accepted. Please revise the paper accordingly based on comments.